# An Update on Australian Policy around Lamb Marking with Examination of Potential Drivers. Comment on Johnston et al. How Well Does Australian Animal Welfare Policy Reflect Scientific Evidence: A Case Study Approach Based on Lamb Marking. *Animals* 2023, *13*, 1358

**DOI:** 10.3390/ani14192890

**Published:** 2024-10-08

**Authors:** Charlotte H. Johnston, Amanda J. Errington, Mark R. Hutchinson, Alexandra L. Whittaker

**Affiliations:** 1School of Biomedicine, Faculty of Health and Medical Sciences, University of Adelaide, Adelaide, SA 5000, Australia; charlotte.johnston@adelaide.edu.au (C.H.J.); mark.hutchinson@adelaide.edu.au (M.R.H.); 2EthiQualia, Melbourne, VIC 3000, Australia; mandy@ethiqualia.com.au; 3Davies Livestock Research Centre, University of Adelaide, Roseworthy, SA 5371, Australia; 4Australian Research Council Centre of Excellence for Nanoscale BioPhotonics, University of Adelaide, Adelaide, SA 5000, Australia; 5School of Animal and Veterinary Sciences, The University of Adelaide, Roseworthy, SA 5371, Australia

**Keywords:** animal welfare legislation, mulesing, pain relief, sheep, Australia

## Abstract

**Simple Summary:**

In Australia, animal welfare legislation is being reassessed and a number of notable changes have occurred. This article clarifies statements and builds upon a previous article published in 2023, titled “How Well Does Australian Animal Welfare Policy Reflect Scientific Evidence: A Case Study Approach Based on Lamb Marking”. In the article, we explore the debate and changes around mandatory pain relief for mulesing of lambs and how this differs between Australian states and territories. The article also looks at how these legislative changes and industry action influence practice and how the interest of the public is represented in animal welfare law. It stresses the need for clear communication, collaboration among stakeholders, and policies based on scientific evidence to improve animal welfare.

**Abstract:**

This commentary provides an update and clarification on the legislative landscape surrounding mulesing in Australia since the publication of the 2023 study, “How Well Does Australian Animal Welfare Policy Reflect Scientific Evidence: A Case Study Approach Based on Lamb Marking”. The article explores legislative changes mandating the use of pain relief for mulesing in various states, emphasising Victoria’s original role, and highlighting the fragmented state-based legislative approach to animal welfare. It discusses the impact of these legislative changes on industry practices and animal welfare outcomes. The commentary highlights the complexities of policy development in this area, due in part to the diverse and often conflicting interests of stakeholders and the public. It underscores the importance of transparency, stakeholder collaboration, and scientifically informed policymaking to effectively enhance animal welfare standards.

## 1. Introduction

The article, “How Well Does Australian Animal Welfare Policy Reflect Scientific Evidence: A Case Study Approach Based on Lamb Marking”, published in April 2023, discussed the policy surrounding surgical procedures performed at marking [1]. These procedures, in the Australian context, involve a range of tissue modifications including castration, tail docking, and mulesing. We would like to focus on mulesing to clarify comments made in the article and provide an update on recent changes in legislation. This is a rapidly evolving area and, as such, attracts attention from a variety of parties, each influenced by local and global factors, some of which are steeped in tradition and slow to change, while others flow more freely, responding to volatile factors such as activist-led marketing campaigns and social media. Balancing the interest of the animals themselves, stakeholders, and the community at a local and international level is exceptionally challenging and made more difficult by the fragmented state-based legislative approach in Australia. 

This article is framed as a comment on the previous article. Firstly, the article clarifies the legal position regarding mulesing in the state of Victoria. Recent legislative changes in other jurisdictions are then highlighted, with some discussion on the impact of these changes on industry practice. A discussion of industry and public sentiment surrounding mulesing and the implications of mandating and enforcing the provision of pain relief follows. Finally, we discuss how this relates to the sustainability of the livestock industry.

## 2. Clarification

In the original article, there were two mentions of the voluntary nature of the Code of Accepted Farming Practice for the Welfare of Sheep(Victoria)(Revision Number 3) [2]. As it stands, this, in itself, is correct; however, the example given that pain relief for mulesing is a requirement under the code but that there may not be any direct ramifications for failing to comply requires further clarification. The statements from the original article are listed below:

Page 10. “Victorian guidelines also require analgesia for all sheep, plus anaesthetic if over 6 months, although it is important to remember that this is a voluntary code, so this may not actually be the routine practice in this state.” 

Page 17. “Vic provides an excellent example of this with a Code that arguably is the most welfare friendly by requiring all mulesed animals to have pain relief and for operators to be formally trained in the procedure. However, this code is a voluntary code of practice in this state, and therefore, there are no direct ramifications for failing to adhere to these provisions.” 

Victoria was the first state to pass legislation mandating the use of pain relief for mulesing. The *Prevention of Cruelty to Animals Regulations 2019* (Vic) [3] issued under the *Prevention of Cruelty to Animals Act 1986* (Vic) [4] (Figure 1), state that any sheep to be mulesed must be “administered with a pain-relief product that has been registered for use on sheep by the Australian Pesticides and Veterinary Medicines Authority” (APVMA) (r. 8(2)). This legislation came into operation in July 2020, following the sunsetting of the previous regulation. Therefore, in the sole case of mulesing, the provision of pain relief is compulsory, and non-compliance is an offence. The penalty for non-compliance is 20 penalty units, equating to AUD 3846.20 from 1 July 2023 to 30 June 2024, based on unit values provided in the *Monetary Units Act 2004* (Vic). There is also an infringement penalty of 3 units, equating to an on-the-spot fine of up to AUD 577 (*Monetary Units Act 2004* (Vic) listed in sch. 5 of the Regulations) [5]. Penalty units are adjusted annually. We would like to take this opportunity to explore this statement and provide further updates in this rapidly evolving domain. 

## 3. Legislative Changes Regarding Mulesing across Australia

In Victoria, the *Prevention of Cruelty to Animals Regulations 2008* (Vic) were due to expire in December 2019, after the conclusion of their 10-year duration with an additional 12-month extension [7]. This provided the opportunity for review and inclusion of the requirement for pain relief during the mulesing procedure (Table 1), which came into effect 6 months after the regulations came into force. Specific references to mulesing in all state and territory regulations are depicted in Table 1. 

In New South Wales, there was a parliamentary inquiry into the *Prevention of Cruelty to Animals Amendment (Restrictions on Stock Animal Procedures) Bill 2019* (NSW) initially introduced to the Legislative Council by Hon Mark Pearson MLC of the Animal Justice Party in August 2019 [8]. The bill proposed mandating pain relief for mulesing prior to a complete ban by January 2022 and additionally mandating pain relief for other husbandry procedures including tail docking, castration, and ear tagging. The final report recommended that the bill not pass in its current form citing the efficacy of the mulesing procedure for preventing flystrike and the lack of viable alternatives. Moreover, the short time frame of the ban was deemed insufficient to allow smooth transition without detrimental welfare outcomes and substantial industry losses. Mandating pain relief for other husbandry procedures across multiple species was also not endorsed due to concerns predominantly focused on withholding periods associated with providing analgesia for procedures like ear tagging [9]. Ear tagging may be performed immediately prior to transportation for identification in cases where the original ear tag failed or as part of farm protocol, so if any product with a withholding period was used at that time, it would delay transportation or risk exceeding maximum residue limits if the animal were sent to slaughter [10]. The high level of voluntary industry uptake of pain relief for mulesing was also considered as adequate evidence that legislative change was unnecessary, opting instead for industry-led change [10]. 

Following the failure of the bill to pass into legislation, Hon Mark Pearson MLC presented a petition signed by more than 2800 citizens requesting a phasing out of mulesing by 2030 and an immediate mandate for pain relief for mulesing, castration, and tail docking in September 2022. The government elected, again, not to make any of the changes, stating support for industry-led initiatives and nationally consistent standards and guidelines, such as the Australian Animal Welfare Standards and Guidelines (AAWSG) for Sheep [11,12].

More recently, Tasmania passed legislation reforming the *Animal Welfare (Sheep) Regulations 2023* (Tas) to mandate appropriate pain relief be administered to any lamb being mulesed [13] (Table 1). The term “appropriate pain relief” used in the Tasmanian regulations is notable in that it differs from the Victorian phrasing, simply requiring that a registered pain-relief product be used. This deviation in phrasing may reflect the debate around the efficacy, and, thus, “appropriateness”, of registered analgesics for the mulesing procedure [14,15]. During consultation regarding the new *Prevention of Cruelty to Animals Regulations 2019* (Vic), there were 816 comments referring to mulesing in submissions from stakeholders and interested individuals or groups. There were concerns raised about the timing and duration of pain relief before and after the procedure in 42% of these comments, with only 5% stating that pain relief is unnecessary or too expensive [15]. Ultimately, the difference in wording is unlikely to change the practical outcome of the mandates and, realistically, they affect only a small proportion of producers not already using pain relief for mulesing [16]. This small population of producers and the registered pain-relief options available are discussed further in Section 4. 

There were numerous submissions to the various bodies responsible for assessing the changes in law surrounding mulesing practice mentioned above. Across these submissions, there were common themes; namely, animal welfare-associated organisations and many public submissions focused on the brutality of the procedure, calling for an immediate ban or phase out with the mandating of pain relief in the interim period [17,18,19]. Alternatively, industry groups argued that the quick, one-off procedure provides lifelong protection against flystrike, a devastating disease that inflicts immense and prolonged suffering with high rates of mortality [9,20]. In Victoria and Tasmania, the final decision not to ban the procedure, but rather mandate pain relief, came from the lack of viable alternatives to mulesing and the severe compromise in welfare from breech flystrike [7]. There was industry support for pain relief to be mandated in many cases, although not unanimously [9,20,21,22]. The *Prevention of Cruelty to Animals Amendment (Restrictions on Stock Animal Procedures) Bill 2019* (NSW) [8] sought to mandate pain relief for several routine husbandry procedures across farmed species, attracting concerns surrounding withholding periods and the practicalities of providing pain relief for procedures such as ear tagging, for which many deemed pain relief unnecessary [10]. Furthermore, there was concern that pain relief might by detrimental to animal welfare due to potential limitations on the transporting of animals as a result of withholding periods. It is possible that this stretch to mandate pain relief beyond mulesing prevented the bill being passed in any form and highlights the role of collaboration and compromise between industry and animal welfare organisations in creating policy change. animals-14-02890-t001_Table 1Table 1State and territory animal welfare regulations and parts specifically relating to mulesing.JurisdictionRegulationContents Relating to MulesingACT*Animal Welfare Regulation 2001* [23]No specific reference to mulesing.NSW*Prevention of Cruelty to Animals Regulation 2012* [24]No specific reference to mulesing.SA*Animal Welfare Regulations 2012* [25]“Part 9—Sheep79—Mulesing1)A person must not carry out the Mules operation on a sheep that is less than 24 h old or more than 12 months of age.2)A person carrying out the Mules operation on a sheep must not remove any skin from the sheep unless it is wool-bearing skin.3)A person who has the care, control and management of a sheep must ensure that the
a)Mules operation is not carried out on the sheep unless—
i)it has good body condition; andii)if the sheep is over 6 months of age—it is given a suitable analgesic or is anaesthetised.”
QLD*Animal Care and Protection Regulation 2023* [26]“Schedule 3 Code of practice about sheepPart 2—Husbandry proceduresDivision 2 Carrying out husbandry procedures8 Mulesing sheep1)Subject to subsections (2) and (3), a person may perform mulesing on sheep if the sheep are more than 24 h old but not more than 12 months of age.2)A person performing mulesing on a sheep must ensure—
a)the mulesing is done in a way that—
i)avoids unnecessary pain and suffering for the sheep; andii)does not remove wool-bearing skin from the sheep; andb)if the sheep are more than 6 months of age—suitable pain relief is administered to the sheep.
3)A person must not perform mulesing on a sheep if the sheep is showing signs of disease, weakness, illness or injury.”TAS*Animal Welfare (Sheep) Regulations 2023* [13]“Part 3—Procedures and treatment performed on sheep14. General requirements for procedures and treatment“A person performing [mulesing] must ensure that the sheep does not feel unreasonable pain, distress, suffering or injury during, or after, the procedure”19. Mulesing1)A person performing mulesing of a sheep must —a)have the relevant knowledge, experience and skills to safely perform mulesing of the sheep, or be under the direct supervision of a person who has such knowledge, experience and skills; andb)use appropriate tools and methods to undertake mulesing of the sheep.
2)A person who is mulesing a sheep must —
a)not perform the procedure on a sheep that is less than 14 days old or more than 6 months old; andb)give the sheep appropriate pain relief for the procedure; andi)only remove wool-bearing skin when mulesing the sheep.3)A person must not mules a sheep that shows signs of a debilitating disease, weakness or ill-thrift.4)The owner of a sheep must ensure that the sheep is not mulesed while the sheep shows signs of a debilitating disease, weakness or ill-thrift.”VIC*Prevention of Cruelty to Animals Regulations 2019* [3]“Part 2—Protection of AnimalsDivision 1 General Requirements8 Sheep1)A person must not mules a sheep unless the sheep is administered with a pain-relief product that has been registered for use on sheep by the Australian Pesticides and Veterinary Medicines Authority.”WA*Animal Welfare (General) Regulations 2003* [27]No specific reference to mulesing.NT*Animal Protection Regulations 2022* [28]No specific reference to mulesing. 


## 4. Impact of Legislative Changes 

In this section, we will discuss how the changes in regulation influence the provision of pain relief at mulesing and the overall awareness of animal welfare legislation. Additionally, the impact of these changes will, in part, involve a level of enforcement, which is also discussed. 

### 4.1. The Use of Pain Relief for Mulesing

Accurately determining the actual use of pain relief on a farm can be challenging. Industry funded surveys of producers are often used to track practices and measure attitudes to inform the future use of levy derived funds. As part of a broader survey for the Sheep Sustainability Framework supported by Meat and Livestock Australia (MLA) and Australian Wool Innovation (AWI), 1203 Merino producers were asked about their 2021 husbandry practices via online surveys and telephone interviews. The survey reported that 100% of the 264 surveyed Victorian producers that mules used pain relief, with roughly 9% using a combination of anaesthetic spray and analgesic injection or analgesic oral gel [16]. An earlier AWI survey, assessing husbandry practices in 2017, reported that 85% of Victorian producers that mulesed used pain relief [29]. In NSW, where mandating pain relief was subject to a parliamentary inquiry in 2019 but did not progress, the 2017 survey reported that 88% of producers were using pain relief, and this increased to 93% of the 409 NSW producers interviewed in the 2021 survey. Based on this survey, it would appear that the pain-relief mandate was successful in achieving a minimum standard of pain relief for all lambs mulesed in Victoria. However, there are limitations to any survey: only 5% of Australian Merino producers were surveyed in 2021, and the respondents were sourced from the Meat and Livestock Australia (MLA) member database [16]. Balanced sampling across states and flock size was used to better reflect the population [16], but even so, the producers’ responses were not externally verified and there may be a level of bias, particularly when answering a question that would implicate the respondent in committing an offence. 

Another point raised in the survey by Colvin et al. [16] was the reason for not using pain relief for mulesing, with the majority of the respondents not using pain relief either stating that it was not necessary (35%), or they had not considered it or had no reason (22%). From the 50 respondents that reported not using pain relief for mulesing, some reported confusion about how the pain relief would work with flystrike preventative products and others stated that the “contractor didn’t have time/said it was a hinderance”. Also, of the producers still mulesing, the majority (60%) were unlikely or very unlikely to cease mulesing [16]. While it is not possible to determine if the uptake of analgesia in Victoria was because of the change in legislation, the findings of the survey suggest that without mandating pain relief there will always be a small proportion of producers that will not use pain relief despite industry efforts to promote the use of analgesia for surgical husbandry procedures [30]. 

Animal welfare legislation is typically intended to reflect minimum standards of care acceptable to society [31], and its substance considers prevailing economic and political influences, including those from industry stakeholders, animal advocacy groups, and other parties [1,31]. Policymakers generally introduce laws or mandates to clarify existing social norms that are perceived as morally correct [32]. Introducing a mandate to enforce the provision of pain relief for one specific surgical husbandry procedure after decades of debate and research into viable alternatives is a small step in improving animal welfare and protection through legislation. Additionally, given the widespread industry uptake of pain relief already, the real animal welfare benefits are minimal. Nonetheless, the change in legislation marks a large shift in industry and public attitude. It also shows recognition that mulesing has become a significant social concern and there is a need to support industry in upholding a minimum standard. 

In theory, the intention of mandating the provision of pain relief is to reduce the suffering of all lambs undergoing the mulesing procedure. While the type of pain relief is easily described, the result of giving that pain relief to the individual animal is harder to quantify in order to establish its effectiveness in reducing pain. Firstly, not all pain relief is equal. Most farmers use the local anaesthetic product, Tri-Solfen [16]. This product numbs the breech area for a number of hours following the mulesing procedure. It is easy to apply and has been classified as a Schedule 5 agent, meaning it is available to producers without a veterinary prescription. Given the ease of access and use, it is likely that the mandate will result in farmers that were not using any pain relief adopting Tri-Solfen. While there is a reduction in pain, it does not obliterate the suffering experienced by the lambs during the surgical procedure or after the product wears off, it also does not completely stop the pain while active. Scientific evidence suggests that a combination of non-steroidal anti-inflammatory drug (NSAID) pain relief and Tri-Solfen is more effective at reducing the pain from mulesing compared to either agent alone [33,34]. Unfortunately, there is little incentive to adopt this approach as it is more expensive and time-consuming to the producer and there is limited market reward for providing this level of pain relief. The accessibility of injectable NSAIDs is also comparatively limited by the requirement for a prescription. However, a product combining a 1% concentration of meloxicam with a combination vaccine typically administered at marking has been amended to Schedule 6 by the Therapeutic Goods Administration (TGA) [35], but is yet to be assessed and registered by the APVMA. This product would allow the administration of an NSAID without the requirement for a veterinary prescription or the need for additional handling or drug administration time and so may improve producer uptake, but it is currently not available as it has not been fully tested and approved by the APVMA. Buccalgesic, an oral form of meloxicam, is already available to farmers under Schedule 6, although uptake of this product is poor—it is used by only 5% of producers who use pain relief for mulesing [16]. The additional benefit of using an NSAID is that pain relief is simultaneously provided for other painful procedures that occur during marking, such as castration. Despite this, even with the use of NSAIDs and local anaesthetic, which provide a degree of pain relief for around 24–72 h [36], the pain caused by mulesing has been shown to last for at least 2 weeks [37,38], with full wound healing occurring four to seven weeks post procedure [39,40]. There are no registered analgesic agents for sheep that would provide a degree of pain relief for that duration or for the procedure itself [41]. 

Understanding the motivations behind adopting pain relief for mulesing is important for the promotion of these practices among all producers and to aid in developing innovative and practical solutions. In the 2021 survey [16], Merino producers that used pain relief for mulesing were asked to list their reasons for using particular forms of pain relief. There were 649 producers using Tri-Solfen, 29 using meloxicam injection, and 38 using meloxicam oral gel (Buccalgesic). Most producers listed reasons related to product effectiveness and animal welfare benefits. Of the producers using meloxicam products, 73% reported that they used them because they worked to reduce pain, whereas only 56% of respondents reported that they used Tri-Solfen because it reduced pain. Respondents generally found meloxicam injection (43%) easy to apply, followed by Tri-Solfen (38%), then Buccalgesic (26%). No respondents reported that Buccalgesic had been recommended by a retailer, contractor, or stock agent, compared to 11% for Tri-Solfen, and 8% for the meloxicam injection. Nine percent of respondents listed veterinary recommendation as a reason for using a form of pain relief. Only 16% of respondents reported that they used pain-relief products because it was industry standard. Based on these findings, the major driver for using pain relief appears to be the care for animal welfare and reducing pain. This is consistent with other research indicating that farmers care about minimising pain associated with invasive husbandry procedures [42,43], although it does not clearly indicate why so few producers are using combinations of pain relief, which is known to be best practice, or why the vast majority of producers choose Tri-Solfen over meloxicam products if they only select one form of pain relief. Retailers, industry, and veterinarians seem to have a relatively small influence over these decisions; perhaps if there was greater attention focused on improving relations and communication between these groups and producers, more producers would be using combination pain relief. It also reveals a possible reason why there is a slow move away from breeding plain-bodied sheep, as it would appear that veterinarians and industry groups are not playing a large role in the decision-making process around mulesing. If we look to the producers not using pain relief, many of them do not see a need for administering pain relief, as discussed above. This attitude is more pronounced in relation to castration and tail docking procedures. In the National Producer Survey in 2022, published in the Sheep Sustainability Framework [44], 75% of producers do not use pain relief for castration and 56% do not use pain relief for tail docking. Of these, 45% and 50% of producers felt that pain relief was not necessary for castration and tail docking, respectively. Less than 10% of those producers listed cost as a barrier and roughly a quarter had not considered using pain relief.

### 4.2. Beyond Pain Relief for Mulesing

Bringing components of the Animal Welfare Standards and Guidelines into regulations, and, thus, making violations punishable by law, may also improve awareness of the Standards through enforcement and wider dissemination of the changes to legislation. The AWI survey in 2021 reported that 86% of Merino producers were aware of the Australian Animal Welfare Standards and Guidelines for sheep, although only 57% of those had read them, and 14% had not heard of them. Just over 10% of respondents from each state reported that they had not heard of the Standards, even in those states where the Standards are mandatory [16]. Industry actions to improve awareness of animal welfare legislation include communication with producers through local and national news and media outlets, educational communications from levy-funded industry bodies, and assurance schemes that require members to be aware of, and have read, the Standards, for example, MLA’s Integrity Systems Livestock Production Assurance (LPA) program (https://www.integritysystems.com.au/on-farm-assurance/animal-welfare/ accessed on 7 August 2024) and Better Choices (https://www.betterchoices.com.au/industry/farmers-and-producers/ accessed on 7 August 2024). 

### 4.3. Enforcement of Legislation

When it comes to implementing animal welfare reform through legislation, enforcing any changes in law is vital to the legitimacy and efficacy of that reform. However, enforcing animal welfare law is difficult and prone to challenges including insufficient resourcing, and conflicts of interest among enforcing bodies. Consequently, there is ongoing debate surrounding the practicalities of enforcing animal welfare laws and their purpose [45]. 

Currently, the major agencies responsible for enforcing animal welfare laws in livestock are the state and territory government departments of primary industries and agriculture (Table 2). Even where non-government organisations are the predominant enforcers, they may delegate to government departments or not be effectively resourced for investigations or monitoring [45,46]. For example, there are only seven inspectors for the Royal Society for the Prevention of Cruelty to Animals (RSPCA) in South Australia, five of who operate across the state. They respond to over 4000 reports of animal cruelty each year and, additionally, are involved in visits to sale yards, rodeos, abattoirs, and feedlots [47]. In South Australia, where the majority of animal law enforcement is performed by the RSPCA, only 25% of closed animal welfare offences from 2006 to 2018 involved livestock [48]. The South Australian RSPCA inspectorate are fully funded by the South Australian Government, although in most of the states and territories the RSPCA inspectorates rely on community donations to support state government funding of their activities, such as in Victoria, NSW, and Queensland [49,50,51]. The discrepancies between which enforcement agencies cover commercial livestock welfare and funding arrangements creates greater confusion around how to report incidences of cruelty and how compliance monitoring is conducted. Due to limitations in staffing and funding, enforcement agencies tend to take a more reactive role, responding to reports of cruelty to animals, with less focus on auditing and monitoring compliance [46]. Where compliance checks do occur, they tend to be in areas of livestock congregation, such as rodeos, saleyards, abattoirs, and feedlots [47]. This is a targeted use of limited resources to monitor a large number of animals from a range of properties in a relatively confined space and timeframe. The existing limitations in animal welfare policy enforcement have been raised by several of the recent Animal Welfare Legislation reviews conducted in various states [46,52,53,54]. 

Co-regulation, where industry arrangements are used as a form of regulation, has been discussed through a number of the recent animal welfare act reviews as a means of reducing the regulatory burden on governments and non-government organisations [53,65]. An example of co-regulation is self-reporting through industry-led quality assurance programs, some of which require regular on-farm audits for higher levels of certification [66]. There appears to be some support for this method as a part of a larger regulatory compliance mechanism that has independent oversight separate from industry to avoid conflicts of interest.

To monitor strict compliance to the new regulations on mulesing in Victoria and Tasmania, on-farm visits at the time of lamb marking would be required. Logistically, this would be very difficult and extremely costly. Instead of this, most animal welfare enforcement agencies focus on the promotion of legislative objectives and education of the public, industry, stakeholders, and owners suspected of animal mistreatment. These agencies tend to reserve prosecution for serious cases of cruelty, in part due to the high associated costs, and to use limited resources efficiently [47,49,63]. The Department of Primary Industries and Regional Development for Western Australia (DPIRD) has clearly laid out this regulatory approach in a recent report where they discuss the implementation of strategies to improve attitude to compliance, thus, reducing compliance costs, as depicted in Figure 2, which was adapted from the Australian Taxation Office Compliance Model [63]. 

Industry action and funding have been instrumental in incentivising the use of pain relief at mulesing through aiding in development, advocacy, and the promotion of the use of registered medications that are both easy and efficient to administer and obtain. There are also financial incentives on wool sales for those using pain relief at mulesing and for those ceasing mulesing [66,67]. Additionally, there is substantial industry investment into education and promotion of the use of pain relief for other painful husbandry procedures and for the phasing out of mulesing [44,68]. In relation to the compliance model in Figure 2, the two lower categories may already be effectively “saturated” as a result of the actions listed above. Consequently, it is possible that those producers or contractors not already using pain relief may lie at the upper half of the pyramid, that is, they do not want to provide pain relief and/or they have decided not to comply. In that case, the method of ensuring compliance, based on this model, would be to use monitoring and inspection, and to impose penalties where necessary. It is possible that the introduction of mandatory regulations themselves provides sufficient downward pressure to influence this population of producers. If the producer surveys discussed earlier truly reflect the changes in Victoria following mandating pain relief, it would appear that the law has effectively encouraged those producers unpersuaded by industry and market pressure to use pain relief for mulesing. However, without consistent compliance monitoring it is difficult to know if this is the case. Future research on this issue would provide valuable insights for policymakers developing farm animal welfare legislation. Optimising farm animal welfare through legislative means would require a higher standard of enforcement and compliance monitoring. Typically, the law is not designed to be at the forefront of social change but rather to maintain a minimum standard that is acceptable to the community.

Understanding community expectations of acceptable welfare for farm animals is important for the development of welfare legislation and for analysing its regulatory impact [46,69]. The ideal that good animal welfare is closely related to increased productivity is given as a reason for the strong leadership of industry stakeholders in the development of farm animal welfare law [46]. While productivity and welfare are linked, they do not necessarily go hand-in-hand and the interpretation of good animal welfare has been shown to differ between producers, animal welfare scientists, animal rights activists, policymakers, and the public [70,71]. This disconnect has been demonstrated during incidences of rapid increases in public advocacy and media attention surrounding alleged farm animal welfare violations that exist within the current legislation [30,72,73]. For example, global scrutiny and increased market pressure in the mid-2000s pushed for Australia to ban the legal practice of mulesing. While this pressure resulted in significant investment into research and education to support alternatives to the practice, legislative change has been slow, with only two states having mandated the use of pain relief for mulesing [3,13] and 52% of Australian Merino woolgrowers still mulesing their ewe lambs [16] almost two decades later.

Concerns about conflicts of interest in farm animal welfare regulation were raised in the 2016 Productivity Commission Inquiry Report on the Regulation of Australian Agriculture [46]. Conflicts of interest can arise when government agencies charged with administering animal welfare law are also advocates for the industry’s interests; this is often referred to as regulatory capture. The distortion of public interest by industry can result in policies favouring industry interests over community expectations of animal welfare [69]. The Productivity Commission report made recommendations to improve animal welfare regulation and limit the risk of regulatory capture by establishing an independent committee for Animal Welfare consisting of members appointed based on their skills and experience, not as representatives of industry, government departments, or animal welfare organisations [46]. Support for independent animal welfare advisory committees at federal and state level has been noted in reviews of animal welfare legislation in Victoria [74], South Australia [65], New South Wales [52], and Western Australia [54]. 

## 5. Industry and Public Sentiment Surrounding Mulesing 

Mulesing was introduced in the 1930s and rapidly became popular among Merino producers to reduce the risk of breech flystrike. While there were some concerns raised regarding the impact of the procedure on lambs, the lifetime protection against flystrike was seen as a worthy compromise. With growing understanding of animal sentience and public concern related to the procedure, the wool industry has responded by promoting breeding initiatives to produce plainer bodied sheep that do not require mulesing and other mulesing alternatives such as regular crutching and the use of insecticides. The industry committed to a ban of mulesing in 2010, but, due to the lack of viable alternatives, had to rescind on that commitment [75]. Australia is the only country that continues to use mulesing as a form a flystrike prevention. New Zealand officially banned the procedure in 2018 after gradually phasing out the practice, largely through industry led initiatives following pressure from animal welfare groups and markets in 2004. The vast majority of NZ farmers had voluntarily ceased mulesing by 2010, well before the legislation to ban mulesing was introduced [68]. The market dominance of a single wool broker, the New Zealand Merino Company, created significant pressure on the New Zealand wool industry to cease mulesing due to the company’s strong advocacy for non-mulesed wool. Additionally, the attitude towards mulesing within the New Zealand farming population was that mulesing was more of an Australian practice. The difficulty in phasing out mulesing in Australia lies in its climate, greater proportion of Merinos, slower move to breeding less wrinkled sheep, larger enterprises, labour costs and availability, chemical resistance, cultural norms and attitudes, and different wool industry [68,76]. In Australia, the stronger cultural practice of breeding wrinkled Merinos and mulesing, the perceived difficulty and expense of transitioning to a non-mulesed flock, and a more diverse wool industry have created less pressure on producers to adopt management practices which would reduce the need for mulesing [68,77].

Public concern surrounding farm animal welfare is increasing and, consequently, as discussed above, both government and industry groups are responding by actively engaging in consultation and introducing new laws and industry incentives to address the shift in social values. However, there is ongoing discourse as to whether these actions sufficiently reflect the public’s level of concern [76]. While pain relief for mulesing has been mandated in two states, which was seen as a suitable compromise, there are still calls from animal welfare groups, international markets [78], some industry stakeholders, and members of the public to ban the practice of mulesing. The Australian Alliance for Animals, Four Paws Australia, and Humane Society International Australia released a report in September 2024 marking 20 years since the original commitment by the Australian wool industry to ban mulesing. The report outlines the history of mulesing, the growing scrutiny of the practice, and the reasons behind the limited uptake of genetic selection for plain-bodied sheep, and it calls for greater government action to implement a ban [76]. Recent reviews of Animal Welfare Acts across a number of states have highlighted a growing concern surrounding mulesing and other surgical husbandry procedures during community consultation processes [54,65,74]. There is also a clear separation in submissions from members of the public and animal welfare organisations in comparison to representatives of the livestock industry. Animal welfare groups and the public tend to feel that the current list of prohibited procedures is insufficient and often list mulesing as a procedure that should be banned. As discussed briefly in the previous section, the term animal welfare is interpreted differently by different groups. This is evident across a variety of research showing a disparity between the public’s ideas of what good animal welfare is and farmers’ ideas [75,79,80]. For example, comparison of the perceived pain intensity of husbandry procedures between the general public and farmers revealed a divide between the groups, with farmers typically perceiving procedures as less painful than the public [81,82]. This separation of public sentiment and farmer perception may derive from differing levels of exposure and understanding of the procedure and the consequences of not performing it [42,43]. A lack of education and understanding around farming systems is contributing to a growing divide between urban and rural communities that is reportedly fuelling public concern around animal welfare [42,43,83]. Some producers suggest that improvements in education will reduce the urban–rural divide and assure the public that farmers are the best advocates for animal welfare. However, this is not always the case, particularly in topics where opinions are often grounded in core values and beliefs, such as veganism [42]. Phillips et al. [79] also suggested that a possible contribution to the growing urban–rural divide is that producers have become inured to injurious procedures, including mulesing, through repeated exposure, but see longer term welfare concerns, such as malnutrition and parasitism, as a greater factor driving the economic productivity of a farm. A 2008 survey of 22 sheep producers from Western Australia indicated that the majority of farmers (16/22) disliked mulesing but the short-term pain was outweighed by the long-term protection against flystrike. A few of the farmers also elaborated on the difficulty of tending to flyblown sheep, including the prolonged suffering of the animal, particularly those animals that succumb to sepsis or are euthanised [75]. Mulesing, to the uninitiated (i.e., the general public), is a visually gruesome procedure that lasts for less than a minute, meaning it lends well to sensational media releases and advocacy campaigns, whereas capturing the suffering of flystrike is not as immediately captivating. The marketing strategy by PETA in 2004 effectively utilised graphic images of the mulesing procedure to mount a wave of public concern nationally and internationally, succeeding in driving global brands to boycott Australian wool [72]. 

Large-scale campaigns by animal rights activists, leading to public outcry and altered consumer behaviour, have typically been the driving force behind substantial improvements in farm animal welfare and policy [31,72,84]. One such example started with the book Animal Machines, written by animal welfare activist Ruth Harrison, and first published in 1964 [85]. Animal Machines described some of the realities of “factory farming” at the time and triggered intense public reaction, prompting the English Parliament to convene the Brambell Committee in order to review intensive farming practices and define animal welfare [86,87]. The Brambell Report of 1965 set out basic animal freedoms and acknowledged the importance of an animal’s affective state [87]. This was later formalised by the Farm Animal Welfare Council (FAWC) as the Five Freedoms and is used extensively in animal welfare policy. The Five Freedoms are: freedom from hunger and thirst, freedom from discomfort, freedom for injury and disease, freedom from pain, freedom from fear and distress, and freedom to express normal behaviour [88]. Highly publicised animal welfare crises often draw a sudden focus of public attention and can create a political window of opportunity where legislative change can occur relatively quickly [30]. Collaboration between industry groups, scientists, stakeholders, non-government organisations, and relevant government bodies that stems from such crises tends to bring about tangible change in practice and policy, albeit often on a smaller scale than intended by the original activism campaign [72]. 

With continued public scrutiny of farm animal welfare and scientific progress, the theory of productivity being synonymous with good animal welfare is gradually evolving. Traditionally, acceptable animal welfare was based on the absence of negative indicators such as disease and neglect. While these factors are important, they project a narrow view of welfare, focusing on protection from cruelty and unnecessary suffering. Now there is a shift in thinking towards positive welfare to give animals a life free from cruelty and a good life that is worth living [88]. 

## 6. Ensuring a Sustainable Industry

The industry has been a great driver of the uptake of pain relief since the production of economically viable and practical products, although one of the key initial motivators for the development of Tri-Solfen was global activism, leading to significant market pressure following the PETA marketing campaign in the 2000s. The “corporatisation” of PETA’s original message involved international fashion brands openly boycotting Australian wool. The industry mobilised in response to this and funded research into alternatives to mulesing as well as pain-relief options, and worked with producers to phase out mulesing. Prior to this extreme approach, animal welfare advocates had been campaigning for decades to change policy and ban mulesing, with little success [72]. Table 3 clearly shows that there was an increase in producers ceasing mulesing from 2006 onwards, with 50% of those producers ceasing mulesing in the 6 years leading up to the survey [16]. 

A sustainable industry requires social licence to operate [89]. Education and dissemination of balanced information is needed to encourage constructive discussion between invested parties based on facts and a real understanding of the issues presented by any proposed solution. Promoting understanding where interested parties may not agree is crucial to enabling constructive policy that represents society. Deliberative techniques to engage community and various stakeholders have been used successfully in planning and infrastructure to create and implement sustainable policy with long-lasting impact [90]. Similar techniques may also be useful in producing improvements in farm animal welfare through collaboration between industry, community, animal welfare organisations, policy makers, citizen juries, and scientists.

The livestock industry must stay abreast of social concerns and scientific understanding to ensure they remain a driving force behind continued improvement in all areas and, thus, maintain trust among society and government. Historically, the livestock industry has operating with limited scrutiny from the public, in part due to the separation between the animal and the final product [89]. Advances in recording technologies have allowed greater surveillance of several farming practices, with some of these images being released to the public, which has bolstered support for animal rights’ group’s advocacy campaigns [73,89]. There is an argument that, without these actions, several improvements in animal welfare would not have occurred or would have taken longer to occur. 

It is also worth considering the impacts of improving animal welfare on other sectors that influence the sustainability of the industry. For example, banning mulesing without prior selection to reduce breech wrinkle may lead to a greater reliance on chemicals to prevent and treat flystrike [10,68]. Resistance to insecticides and antimicrobials is a major concern for the farming community and the global population currently, and increasing the use of these chemicals may lead to a faster progression of resistance [91]. Additionally, environmental contamination with agricultural chemicals is affecting both local and distant ecosystems [92]. 

A ban on mulesing would also affect the social sustainability of wool production as there would be increased staffing requirements to monitor the flock for signs of flystrike, treat affected sheep, and crutch sheep more frequently to prevent flystrike [68]. Even without a mulesing ban, there is already difficulty finding staff in the wool industry, particularly those who are adequately experienced, to monitor and treat flocks for flystrike. In addition to this, a shortage of veterinarians is greatly affecting rural and regional areas [93], which may be detrimental to animal welfare if a mulesing ban results in an increase in flystrike. 

To create meaningful animal welfare improvements, industry must work collaboratively with animal welfare organisations, government agencies, scientists, and the community to promote a change in response to consumer, market, public, industry, and scientific advancements in order to adapt to the rapidly evolving global market. One such industry initiative is the recently launched Australian Wool Sustainability Scheme (AWSS), building on the current SustainaWOOL Integrity Scheme. This accreditation scheme assesses multiple pillars of sustainable wool production to ensure integrity and traceability in order to maintain global market confidence. Sheep health and wellbeing is one of the main pillars assessed, with the top tier of accreditation requiring that the producer does not mules or has stopped mulesing and uses pain relief for castration and tail docking. The scheme acknowledges the need to move away from mulesing to remain competitive in the global market. As part of a secondary tier of accreditation, growers that still mules are required to provide pain relief for the procedure and are actively supported to plan for a future without mulesing. The wool from these growers is separately certified to ensure full transparency for buyers. Regular audits and annual accreditation confirm farm compliance to the scheme [66]. Initiatives like the AWSS, LPA, and Better Choices often attract motivated producers who are keen on obtaining the best price for their product, are responsive to consumer feedback, and willing to invest in change. While schemes like this can ensure best practice among members, they are not mandatory and, therefore, unable to enforce nationally consistent standards. The 2021 survey reported that only 19% of Merino producers were involved in a quality assurance scheme [16]. 

### The Role of Government in Maintaining a Sustainable Future for Farming

The need for a nationally consistent approach is echoed throughout the numerous submissions, reviews, and reports made to and by various interested parties regarding animal welfare law. The lack of national consistency limits progress in the field of animal welfare due to the threat of introducing competitive disadvantage in states that adopt more “radical” changes to legislation, such as banning mulesing [31,46]. The Animal Welfare Task Group (AWTG) has been created to promote national consistency in animal welfare law specifically for farm animals [94]. This group is made up of representatives from the various state and territory government departments responsible for animal welfare. The AWTG is one of the driving forces working on the renewal of the Australian Animal Welfare Strategy (AAWS), which will provide the framework for nationally consistent animal welfare [95]. The AWTG is also considering reviewing the husbandry practice chapters of the Australian Animal Welfare Standards and Guidelines for Sheep, with the current version being endorsed in 2016, as stated in their 2024 work plan [96]. 

Maintaining strong trade relationships with overseas partners is a key role of government. As seen with the PETA campaign discussed above, pressure from international markets threatened the Australian wool industry. There is growing scrutiny of farm animal welfare globally and a nationally consistent legislative approach in addition to national industry cooperation and collaboration will be a major factor in maintaining a sustainable future export market for Australian wool and sheep products.

Australia is a member country of the Organisation for Economic Co-Operation and Development (OECD). The OECD is an international organisation that develops evidence-based international standards and policies to improve economic performance and social wellbeing globally. In 2023, the OECD updated the OECD Guidelines for Multinational Enterprises on Responsible Business Conduct [97] to include a clause on animal welfare. These guidelines are recommendations addressed to multinational enterprises by governments to encourage positive contributions towards economic, social, and environmental progress. The relevant clause states: 

“Enterprises should respect animal welfare standards that are aligned with the World Organisation for Animal Health (WOAH) Terrestrial Code. An animal experiences good welfare if the animal is healthy, comfortable, well nourished, safe, is not suffering from unpleasant states such as pain, fear and distress, and is able to express behaviours that are important for its physical and mental state. Good animal welfare requires disease prevention and appropriate veterinary care, shelter, management and nutrition, a stimulating and safe environment, humane handling and humane slaughter or killing. In addition, enterprises should adhere to guidance for the transport of live animals developed by relevant international organisations. (Chapter 6, paragraph 85, page 38 [97]).”

While these guidelines are voluntary, claims of corporate misconduct where animal welfare is violated can be brought to National Contact Points (NCP) to be addressed. All countries that are members of the OECD are legally required to establish NCPs for responsible business conduct. These independent offices, which are supported and funded by the government, promote the OECD guidelines and receive and respond to complaints against multinational enterprises that are not meeting the guidelines. The NCP can work with these enterprises to improve conduct and adherence to the OECD guidelines [98]. 

The introduction of animal welfare standards to the OECD guidelines reflects the growing global awareness of the wellbeing of animals in production industries and pertains to the lives of billions of animals. It also places a greater responsibility on OECD member countries to incorporate and update animal welfare standards in their policies [99]. 

## 7. Conclusions

The original article, to which this is a response, discusses the legislation surrounding marking practices in sheep and how scientific evidence shapes policy. It was noted by author AE that the introduction of regulations mandating the use of pain relief for mulesing in Victoria was unclear in the original article. We took the opportunity to address AE’s comments and delve into the discussion around mulesing. This is an area that is rapidly evolving in some respects but remains stagnant in others. Victoria presents an example of where introducing legislation raised the bar for the minimum standard of pain relief, whereas, in NSW, while the debate is active, the legislation relating to mulesing so far remains unchanged.

## Figures and Tables

**Figure 1 animals-14-02890-f001:**
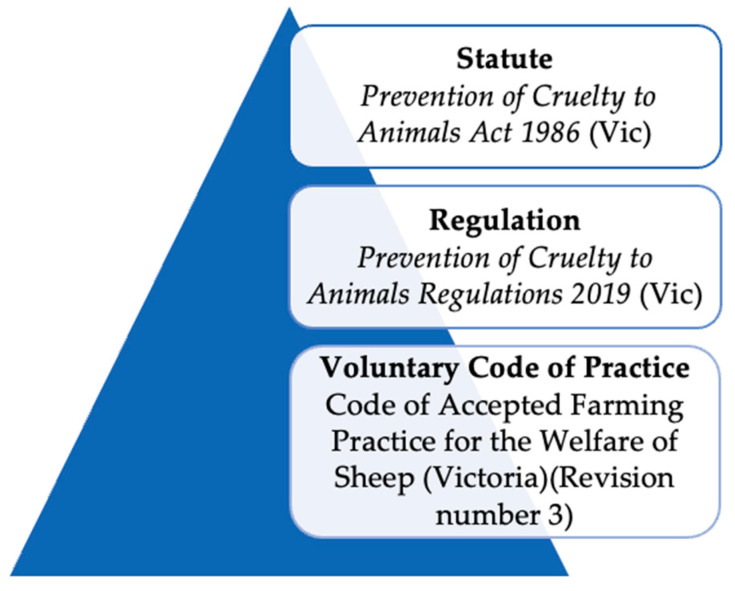
Hierarchy of Victorian sheep welfare protection [2,3,4]. Adapted from Morton et al. [6].

**Figure 2 animals-14-02890-f002:**
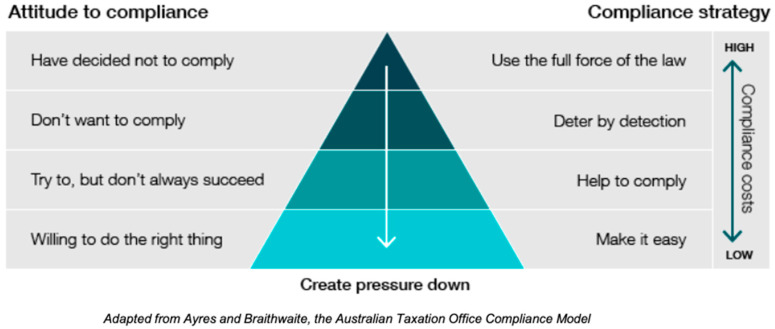
Compliance model demonstrating the relationship between attitude to compliance, the strategy used to improve compliance, and the costs [63].

**Table 2 animals-14-02890-t002:** Predominant livestock animal welfare enforcement agencies in Australia. (Adapted from Morton et al. [45]).

State/Territory	Enforcement Agency ^1^
Australian Capital Territory	RSPCA Australian Capital Territory [55]
New South Wales	RSPCA New South Wales [56] Animal Welfare League NSW [56]
Northern Territory	Department of Industry, Tourism and Trade [55]
Queensland	Biosecurity Queensland [57]
South Australia	RSPCA South Australia [47] Department of Environment and Water [58] Department of Primary Industries and Regions [59]
Tasmania	RSPCA Tasmania [60] Department of Natural Resources and Environment [61]
Victoria	Agriculture Victoria [62] RSPCA Victoria [49]
Western Australia	Livestock Compliance Unit, Department of Primary Industries and Regional Development [63,64] RSPCA Western Australia [64]

^1^ All state and territory police forces are given powers to enforce the animal welfare legislation.

**Table 3 animals-14-02890-t003:** Adapted from Colvin [16]. Proportion of surveyed Merino wool growers that had ceased mulesing by year.

Year Ceased Mulesing	% of Producers (n = 327)
Pre 1991–2000	8
2001–2005	8
2006–2010	17
2011–2015	17
2016–2022	50

## Data Availability

All data are contained within this paper.

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
