# Peer review of "An Update on Australian Policy around Lamb Marking with Examination of Potential Drivers. Comment on Johnston et al. How Well Does Australian Animal Welfare Policy Reflect Scientific Evidence: A Case Study Approach Based on Lamb Marking. Animals 2023, 13, 1358"

_animals, 2024, doi:10.3390/ani14192890_

Round 1

Reviewer 1 Report

Comments and Suggestions for Authors

This article provides interesting commentary in relation to lamb marking and the role of Australian policy in that context. It helpfully identifies why the response to the previous article was considered valueable. 

I had some questions arise as I read that the authors may wish to consider:

- at 234, why is it that only 5% of producers use this product? Perhaps some elaboration on the reasons for uptake might help in this paragraph. 

- at 302-312, Why would unannounced on-farm visits be more difficult and costly than announced ones? Or is it the case that no/minimal visits are made?

- at 334-337, this would seem to be an excellent question for future research, assuming access to producers that have subsequently commenced using pain relief for mulesing, and would provide legislators with excellent information to consider when amending laws relating to farmed animal welfare

- at 385-386, in light of the 2+ decades of pressure in this space, why hasn't greater effort been made to breed less wrinkled sheep and reduce the proportion of Merinos? Why has NZ been so much quicker to respond to these pressures? Is the economic impact significantly different for each country?

- at 407-409, the authors seem to suggest here that if the public had a better understanding of procedures and consequences they would be less opposed to procedures like mulesing. Presumably it could also be argued though that farmers may become desensitised to the pain involved in procedures and also have economic reasons to oppose pain relief. 

Author Response

Dear Reviewer 1,

Thank you very much for reviewing our manuscript and thank you for the kind comments and valuable suggestions.

We have carefully considered the suggestions and made changes detailed below.

Comment 1: at 234, why is it that only 5% of producers use this product? Perhaps some elaboration on the reasons for uptake might help in this paragraph. 

Response 1: Thank you for this comment. We agree and have sought to elaborate where possible. The surveys don’t always give complete clarity around motivations behind responses, so we’ve included some of the reasons given for and against pain relief options, but it doesn’t clearly separate responses. Some of the more in-depth social research in the area has found that producers see the need for providing analgesia for these painful procedures but it doesn’t explore the choices around what to use. More information from the surveys and social science literature has been included on pages 7-8, lines 241-274.

Comment 2: at 302-312, Why would unannounced on-farm visits be more difficult and costly than announced ones? Or is it the case that no/minimal visits are made?

Response 2:  Thank you for pointing this out. We agree with this comment. Currently no regular visits are made as far as I know and the visit would need to take place on the day of marking, which would need to be scheduled with the farmer and/or contractors. We have changed the comment to on-farm visits and excluded the “announcement”, as it doesn’t necessarily change the point, although the purpose of unannounced visits would be to encourage/ensure strict compliance at all times, not just when announced visits occur. This change can be found on page 9, line 336.

Comment 3: at 334-337, this would seem to be an excellent question for future research, assuming access to producers that have subsequently commenced using pain relief for mulesing, and would provide legislators with excellent information to consider when amending laws relating to farmed animal welfare

Response 3:  Thank you for pointing this out, it would be a great question and may even encourage other states and territories to adopt the changes. Please find the updated text on page 10, lines 368-373.

Comment 4: at 385-386, in light of the 2+ decades of pressure in this space, why hasn't greater effort been made to breed less wrinkled sheep and reduce the proportion of Merinos? Why has NZ been so much quicker to respond to these pressures? Is the economic impact significantly different for each country?

Response 4: Thank you for your questions, it is quite a complex issue. We agree that this could use some more detail. Therefore, we have added some more information to the section. These changes can be found on pages 11-12, lines 419-430 and 437-443. 

Comment 5: at 407-409, the authors seem to suggest here that if the public had a better understanding of procedures and consequences they would be less opposed to procedures like mulesing. Presumably it could also be argued though that farmers may become desensitised to the pain involved in procedures and also have economic reasons to oppose pain relief. 

Response 5: Thank you for pointing this out. We agree with this comment. Therefore, we have included further information to clarify our position. This change can be found on page 12, lines 457-467.

Thank you again for all your efforts in reviewing this manuscript. We feel that we have addressed all suggestions and comments in the recently uploaded version.

Reviewer 2 Report

Comments and Suggestions for Authors

This is a well-written manuscript on a very important topic. The authors are to be commended for their comprehensive analysis. 

My few comments for revisions are:

Line 165- "Understanding the true use of pain relief on farm can be difficult to measure." This sentence was a bit awkward. Can you please revise it?

Line 186- "Another point raised by the survey was the reason for not using pain meds..." Please clarify which survey. The previous paragraph referenced several surveys.

Lines 211-212- "While the input, being pain relief, is easily described, the outcome is harder to measure or assure." This sentence is awkward and difficult to understand. Please revise. 

Lines 266-269- Run on sentence. Please check punctuation.

Author Response

Dear Reviewer 2,

Thank you very much for reviewing our manuscript and thank you for the kind comments and valuable suggestions.

We have carefully considered the suggestions and made changes detailed below.

Comment 1: Line 165- "Understanding the true use of pain relief on farm can be difficult to measure." This sentence was a bit awkward. Can you please revise it?

Response 1:  Great point! We agree with this comment. Therefore, we have changed the sentence on page 6, line 163, of the manuscript v2. Updated text: “Accurately determining the actual use of pain relief on farm can be challenging.”

Comment 2: Line 186- "Another point raised by the survey was the reason for not using pain meds..." Please clarify which survey. The previous paragraph referenced several surveys.

Response 2:  Thank you for pointing this out. We agree with this comment. Therefore, we have changed the text slightly to include the reference to the Colvin et al. survey on page 6, line 184. Here is the updated text: “Another point raised in the survey by Colvin et al. [16] was the reason for not using pain relief for mulesing, with the majority of the respondents not using pain relief either stating that it was not necessary (35%) or they had not considered it or had no reason (22%).”

Comment 3: Lines 211-212- "While the input, being pain relief, is easily described, the outcome is harder to measure or assure." This sentence is awkward and difficult to understand. Please revise. 

Response 3:  Thank you for pointing this out. We agree with this comment. Therefore, we have changed the sentence to be less awkward and hopefully easier to understand. The change is on line 209-211 of page 7. Here is the updated text:While the type of pain relief is easily described, the result of giving that pain relief to the individual animal is harder to quantify, in order to establish its effectiveness in reducing the pain experience.”

Comment 4: Lines 266-269- Run on sentence. Please check punctuation.

Response 4:  Thank you for pointing this out. We agree with this comment. Therefore, we have separated the sentence into two as follows:Currently the major agencies responsible for enforcing animal welfare laws in livestock are state and territory government departments of primary industries and agriculture (Table 2). Even where non-government organisations are the predominant enforcers, they may delegate to government departments or not be effectively resourced for investigations or monitoring.” This update can be found on page 8, line 299-303.

Thank you again for all your efforts in reviewing this manuscript. We feel that we have addressed all suggestions and comments in the recently uploaded version.